# Patterns of sex behaviors and factors associated with condomless anal intercourse during the COVID-19 pandemic among men who have sex with men in Hong Kong: A cross-sectional study

**Phoenix K. H. Mo**[1☯]*, **Meiqi Xin**[2☯], **Zixin Wang**[1], **Joseph T. F. Lau**[3,4], **Xinchen Ye**[1], **Kam Hei Hui**[5], **Fuk Yuen Yu**[1], **Ho Hin Lee**[1]

1 Center for Health Behaviours Research, JC School of Public Health and Primary Care, The Chinese University of Hong Kong, Hong Kong SAR, China, 2 Department of Rehabilitation Sciences, Hong Kong Polytechnic University, Hong Kong SAR, China, 3 School of Mental Health, Wenzhou Medical University, Wenzhou, China, 4 School of Public Health, Zhejiang University, Hangzhou, China, 5 Department of Psychology, The Chinese University of Hong Kong, Hong Kong SAR, China

☯ These authors contributed equally to this work.
* phoenix.mo@cuhk.edu.hk

**Data Availability Statement:** Data cannot be shared publicly because of participants' privacy. Data are available from the Chinese University of

## Abstract

### Objectives

The present study examined the patterns of sex behaviors before and during COVID-19, and identified the factors associated with condomless anal intercourse during COVID-19 from individual, interpersonal, and contextual level among men who have sex with men (MSM) in Hong Kong.

### Methods

A cross-sectional study was conducted among MSM in Hong Kong. A total of 463 MSM completed a cross-sectional telephone survey between March 2021 and January 2022.

### Results

Among all participants, the mean number of regular sex partners, non-regular sex partners, and casual sex partners during the COVID-19 period were 1.24, 2.09, and 0.08 respectively. Among those who had sex with regular, non-regular, and casual sex partner during the COVID-19 period, respectively 52.4%, 31.8% and 46.7% reported condomless anal intercourse. Compared to the pre-COVID-19 period, participants reported significantly fewer number of regular and non-regular sex partners during the COVID-19 period. However, a higher level of condomless anal intercourse with all types of sex partners during the COVID-19 period was also observed. Adjusted for significant socio-demographic variables, results from logistic regression analyses revealed that perceived severity of COVID-19 (aOR = 0.72, 95% CI = 0.58, 0.88), COVID-19 risk reduction behaviors in general (aOR = 0.68, 95%

Hong Kong Survey and Behavioral Ethics Committee (contact via sbrec@cuhk.edu.hk) for researchers who meet the criteria for access to confidential data.

**Funding:** The study was supported by the Hong Kong AIDS Trust Fund (MSS 350R). URL of the Hong Kong AIDS Trust Fund: https://www.atf.gov. hk/en/programmes/2020/programmes_mss350r. html The funder had no role in study design, data collection and analysis, decision to publish, or preparation of the manuscript.

**Competing interests:** The authors have declared that no competing interests exist.

CI = 0.48, 0.96), COVID-19 risk reduction behaviors during sex encounters (aOR = 0.45, 95% CI = 0.30, 0.66), condom negotiation (aOR = 0.61, 95% CI = 0.44, 0.86), and collective efficacy (aOR = 0.79, 95% CI = 0.64, 0.98) were protective factors of condomless anal intercourse with any type of sex partners during the COVID-19 period.

## Conclusion

The COVID-19 control measures have caused a dramatic impact on the sexual behavior of MSM in Hong Kong. Interventions that promote condom use during the COVID-19 pandemic are still needed and such interventions could emphasize prevention of both COVID-19 and HIV.

## Introduction

### MSM are facing additional challenges on sexual risk during the COVID-19 pandemic

A rising HIV prevalence has been shown among men who have sex with men (MSM) in Hong Kong [1, 2]. This HIV burden is primarily driven by the high prevalence of condomless anal intercourse [3]. Local reports revealed that 47.9% of MSM had inconsistent condom use [1, 4]. In addition to HIV, MSM are responding to new risks presented by the coronavirus disease (COVID-19). Hong Kong has been facing the fifth wave of COVID-19 since January 2022, it re-emerged and reached the record high with an average of 50,000 confirmed cases per day in March 2022. As of 5 January, 2023, the number of confirmed cases surged to 2,707,624 with 11,936 deaths reported [5]. Hong Kong government has once adopted a number of tightened social distancing and other protective measures to minimise the risk of COVID-19 spreading in the community, e.g. limiting public gatherings to two people, makes wearing masks mandatory in outdoor public places, and closure of catering business such as bars or pubs.

However, studies indicate that COVID-19 can still be detected in semen and feces and persists even after the virus is no longer detected [6–8]. Given the COVID-19 has become a seasonal endemic disease, social distancing and other protective measures may still be applicable. While these measures are considered necessary for disease control, they may have wide reaching consequences. It is essential to understand the impact of COVID-19 on sexual behavior of MSM.

### Changes in sexual behavior during the COVID-19 period

The pandemic has raised concern about how sex behavior will be affected [9]. Some studies from other countries reported that MSM have substantially changed their sexual behaviors and partner selection in response to the pandemic. For example, it has been shown that MSM have reduced the amount of sex and the number of sexual partners during the COVID-19 period [10–13]. They also adopted a variety of strategies to reduce their risk of COVID-19 infection with sex partners, e.g. avoiding certain kinds of partners or activities (e.g., casual partners, group sex parties) [11]. These changes are important not only for reducing the impact of the COVID-19 pandemic, but also for reducing risk of HIV transmission.

On the other hand, other studies have shown that some MSM continued to practice risky sexual behaviors during the COVID-19 period [10]. One study among MSM in Brazil and Portugal found that although majority of them reported a decreased number of sexual partners

and sexual frequency during the sheltering period, over half (53.0%) still had casual sex, 15.8% had group sex, 39.0% had sex under the influence of alcohol or drugs and 30.4% reported condomless anal intercourse [14]. Some men have also increased their use of remote sexual satisfaction options, such as phone sex, webcam sex, and porn consumption [15].

The social distancing measures may also change the partner seeking behavior of MSM. Internet and social and sexual networking apps have long been used by MSM to seek social and sexual connections [16, 17]. The closure of gay bars and saunas leave online sex networking as one of the few options for seeking sexual encounters. Preliminary data has suggested that the use of social and sexual networking has increased during the COVID-19 pandemic [18]. Use of social and sexual networking apps may facilitate higher level of risky sexual behaviors, which pose MSM at higher risk for HIV [16]. A survey among gay, bisexual and transgender men reported that 15% had had sex with a new partner in the week they completed the survey during the COVID-19 pandemic [19].

## Socio-ecological factors of condomless anal intercourse during the COVID-19 period

A broad number of factors at different ecological levels precipitate HIV vulnerabilities. Current studies in this area highlight the importance of utilizing theoretical frameworks to explore multidimensional factors associated with condomless anal intercourse. The Social Ecological approach¸ which proposes that health is explained by multidimensional level of factors, can be a useful framework to understand condomless anal intercourse during the COVID-19 pandemic.

**Factors at the individual level.** HIV-related factors, e.g. risk perceptions, have been regarded as determinants of inconsistent condom use [20]. However, given substantial changes in sexual behavior during the COVID-19 pandemic, factors related to COVID-19 drew more attentions. It is believed that COVID-19 related risk perceptions and perceived impact are important determinants of protective behaviors [21] during the COVID-19 pandemic. Studies among MSM reported that more than half (57%) reported being "extremely" or "very" concerned about COVID-19 [11]. Majority of them agreed that COVID-19 had adverse impacts on their lives such as general wellbeing, social interactions, and drug use and alcohol consumption [10, 14]. The feeling that COVID-19 had high impact on daily life was associated with engaging in casual sex among MSM [14].

**Factors at the interpersonal level.** Social support and social isolation are important protective and risk factors for risky sexual behavior among MSM [22, 23]. The social distancing measures posed in Hong Kong have been unprecedentedly drastic in scale. These measures may have prevented people from receiving social support and limited their sexual networks, which may increase social isolation and exacerbates loneliness and mental problems [19, 24, 25], leading to condomless anal intercourse. Open communication and negotiation on condom use are also the key for of negotiation for safe sex in sex encounter [26]. Availability of condom negotiation script has been emerged as one of the condom use reinforcing elements among MSM in the extant literature [27].

**Factors at the contextual level.** Previous studies have shown that in the MSM community, contextual factors contributed to sexual risk. For example, MSM who have a higher level of collective efficacy are more likely to report positive behaviors [28, 29]. Furthermore, MSM who were frequent gay venue visitors were more likely to engage in high-risk sexual behaviors [30]. Other studies have also found that compared to those recruited from gay venues, MSM who were recruited from the Internet were more likely to engage in condomless anal intercourse and commercial sex, have sought sex partners from the Internet, and have contracted

sexually transmitted diseases [31]. The closure of gay venues or cancellation of social activities due to COVID-19 might have significant impact on MSM's sexual risk.

## The present study

The control measures in response to the COVID-19 pandemic has great impact on sexual behavior of MSM. As the impact of COVID-19 is expected to last, it is important to understand how MSM are adapting their sexual behaviors and their associated factors. No studies to date have explored the multidimensional factors of condomless anal intercourse among MSM during the COVID-19 pandemic. The present study examined the patterns of sex behaviors before and during COVID-19, and identified the factors associated with condomless anal intercourse with any sex partners during COVID-19 period from individual, interpersonal, and contextual level among MSM in Hong Kong.

## Methods

### Participants

Targeted participants were Chinese. Inclusion criteria were: 1) male, 2) aged 16 or above, 3) living in Hong Kong within the past year, 4) able to understand Chinese, and 5) self-reported having had sex with a male.

### Procedure

A cross-sectional study was conducted among MSM in Hong Kong. This study included a convenient sample. Chinese MSM in Hong Kong were recruited through multiple sources. On the one hand, advertisements including information regarding the study and contact information were posted on gay websites in Hong Kong. On the other hand, peer referral was used to further enlarge the sample size. To improve the quality of data collection, interviews were conducted instead of using self-administered questionnaires. Interested individuals contacted a trained and experienced fieldworker to make appointments for the interview. Considering that the Hong Kong government has enacted social distancing measures, telephone interviews were used instead of face-to-face interviews. Prospective participants who were eligible for the study were approached by our trained interviewers through the telephone. Three more follow-up telephone calls were made if prospective participants did not answer the first call. The logistic and purpose of the study were explained, and participants were guaranteed with anonymity and the right to withdraw from the study at any time. Those who agreed to take part provided verbal informed consent, followed by a telephone survey which took about 30 minutes to complete. A HKD $50 supermarket coupon was mailed to the participants for compensating the time they spent. Recruitment and data collection took place between 12 March 2021 and 19 January 2022. Ethics approval was obtained from the Survey and Behavioral Ethics Committee of the Chinese University of Hong Kong.

Out of the 668 MSM who has shown interest to join the study including 460 individuals reached via online advertisements and 208 individuals reached via peer referral, 142 did not answer the telephone calls, 62 rejected to take part, 1 did not meet the inclusion criteria, and 463 met the inclusion criteria and completed the study.

**Sample size calculation.** Local reports showed that around 50% of MSM reported condomless anal intercourse [1, 4]. The sample size of 463 confines 95% confidence intervals of prevalence estimates to about +/-4.5%. Assuming that the prevalence of condomless anal intercourse among those with a potential risk factor will be 60%, the sample size of 463 allows us to detect odds ratios of 1.59 with power of .8 and alpha of .05.

## Measures

**Socio-demographic characteristics.** Socio-demographic variables, such as age, the highest education level, employment status, marital status, and monthly income level were asked.

**Same-sex behaviors before and during the COVID-19 period.** Participants were asked to report the following same-sex behaviors before (i.e. six months before the COVID-19 period) and during the COVID-19 period (i.e. in the past six months). Similar referencing period has been used in previous studies [32].

*Number of sex partners.* Participants were asked to report the number of regular sex partners, non-regular sex partners, and casual sex partners.

*Condomless anal intercourse.* Participants who reported having a sex partner were asked to rate their frequency of condom use in anal sex with regular, non-regular, and casual sex partners on a 3-point Likert Scale from 1 = never to 3 = every time. Participants who score below 3 were classified as having condomless anal intercourse [33, 34].

*Other risky sexual behaviors.* Participants who reported having a sex partner were asked to rate whether they have engaged the following risky sexual practices: having sex that involve more than two people, and having sex under the influence of drugs or alcohol.

*Other sexual behaviors.* Participants were asked to rate whether they have engaged in the following behaviors: masturbation, using sex toys during sex, having phone sex / webcam for sexual satisfaction, and pornography consumption.

**Socio-ecological factors of condomless anal intercourse.** *Factors at the individual level. Perceived risk of COVID-19 infection in general* was measured by 2 items adapted from previous studies [35]. Sample item included "Individuals with COVID-19 are highly contagious even they are asymptomatic". Items were rated on a 5-point Likert Scale from 1 = strongly disagree 5 = strongly agree. The Cronbach's alpha is .68 in the present study. Number and percentage of people who reported agree or strongly agree on each item were also reported.

*Perceived risk of COVID 19 infection during sexual intercourse* was measured by 2 items adapted from previous studies [32]. Sample item included "I am concerned about being infected with COVID-19 from a male regular partner". Items were rated on a 5-point Likert Scale from 1 = strongly disagree 5 = strongly agree. The Cronbach's alpha is .71 in the present study. Number and percentage of people who reported agree and strongly agree on each item were also reported.

*Perceived severity of COVID-19* was measured by a single item. Participants were asked to rate the extent to which they perceive COVID-19 as a dangerous disease on a 5-point Likert Scale from 1 = not serious at all to 5 = very serious [35]. Number and percentage of people who reported agree and strongly agree on this item were also reported.

*COVID-19 risk reduction behaviors in general* was measured by 6 items adapted from previous studies [35]. Such items included adopting social distancing practice (e.g. stay at home, avoid going outside) and personal preventive behaviors (e.g. frequency of washing hands, checking body temperature). Items are rated on a 5-point Likert scale ranging from 1 = never to 5 = always. The Cronbach's alpha is .70 in the present study. Number and percentage of people who reported agree and strongly agree on each item were also reported.

*COVID-19 risk reduction behaviors during sex* was measured by 14 items adapted from previous studies [11, 14]. The items included: avoid having sex, reduce sexual behaviors, wear a mask when having sex, wash hands before and after sex. Items are rated on a 5-point Likert scale ranging from 1 = never to 5 = always. The Cronbach's alpha is .77 in the present study. Number and percentage of people who reported agree and strongly agree on each item were also reported.

*Factors at the interpersonal level. Social support* was measured by six items adapted from previous studies of MSM [36]. Participants were asked to rate the level of emotional, tangible,

and esteem support they can receive from their family and friends on a 10-point Likert Scale from 0 = not at all to 10 = very much. The Cronbach's alpha is .83 in the present study.

*Social isolation* was measured by the 3-item version of the UCLA Loneliness Scale [37]. Items are rated on a 3-point Likert Scale from 1 = hardly ever to 3 = often. The Cronbach's alpha is .84 in the present study. Number and percentage of people of whom the summary score was equal to or higher than 6 were also reported.

*Condom negotiation* was measured the 5-item "Embarrassment about Negotiation and use" Subscale of the UCLA Multidimensional Condom Attitudes Scale [38]. Items were rated on a 7-point Likert Scale from 1 = strongly disagree to 5 = strongly agree. Higher score means higher level of condom negotiation. The Cronbach's alpha is .91 in the present study.

*Factors at the contextual level. Collective efficacy* was measured by a single item. Participants were asked to rate how confident they feel that their community members can practice safe sex behaviors during the COVID-19 period. Item was rated on a 5-point Scale from 0 = not at all confident to 4 = very confident.

*Change in sex seeking environment* was measured by 3 items developed for the study. Participants were asked to rate the level of change in various sex-seeking environment during the COVID-19 period. Items included number of venue where MSM can have sex with a man, number of gay-friendly venues (e.g. bars, saunas), Responses are rated on a 5-point Likert Scale from 1 = greatly reduced to 5 = greatly increased. The Cronbach's alpha is .71 in the present study. Number and percentage of people who reported greatly reduced or reduced on each item were also reported.

### Analysis

Descriptive statistics were presented. Pair-sample t-tests were performed to examine the changes in number of sex partners and Mcnemar tests were performed to examine the changes in other sex behaviors among all participants before and during the COVID-19 pandemic. To identify the individual, interpersonal, and contextual factors of condomless anal intercourse with any sex partners during the COVID-19 pandemic, univariate logistic regressions were first conducted to examine the association between socio-demographic variables and condomless anal intercourse, and the resulting univariate odds ratios (ORu) were presented. Multivariate logistic regression models were then fit for each of the individual, interpersonal, and contextual variables on condomless anal intercourse, adjusted for socio-demographic variables that were significant at the $p < .05$. Resulting adjusted odds ratios (aOR) and 95% CI were reported. All analyses were performed using SPSS Statistics 27.

### Results

### Socio-demographic characteristics of the participants

Socio-demographic characteristics of the participants are presented in Table 1. The majority of the sample were between 18–30 (53.1%) and 31–50 (38.7%) years old. Of the participants, 85.3% were homosexual; 50.1% were single, 84.9% had an educational level of university or above, 68.9% had a full-time job, and 51.4% had a monthly salary of 20,000 HKD or above.

### Pattern and changes of sexual behavior during the COVID-19 period

The pattern and changes of sexual behavior during the COVID-19 period are presented in Table 2. Participants reported an average of 1.24 regular partners (SD = 1.34), 2.09 non-regular partners (SD = 4.58), and 0.08 casual partners (SD = 0.80) during the COVID-19 period. Also, more than half (52.3%) have viewed pornographies, and 47.9% have masturbated during the COVID-19 period. Compared to the pre-COVID-19 period, participants reported significantly

**Table 1. Socio-demographic characteristics of participants (N = 463).**

| | N | % |
|---|---|---|
| **Demographics** | | |
| Age | | |
| 17 or below | 5 | 1.1 |
| 18–30 | 246 | 53.1 |
| 31–50 | 179 | 38.7 |
| 51 or above | 33 | 7.1 |
| Sexual orientation | | |
| Homosexual | 395 | 85.3 |
| Heterosexual | 51 | 11.0 |
| Bisexual | 17 | 3.7 |
| Marital status | | |
| Single | 232 | 50.1 |
| Living with a male partner/married with a male partner/having a male partner but not living together | 228 | 49.3 |
| Others (living with a woman/married with a woman/divorced or widowed with a woman) | 3 | 0.6 |
| Education level | | |
| Secondary or below | 66 | 14.3 |
| University or above | 393 | 84.9 |
| Others | 4 | 0.9 |
| Employment status | | |
| Full-time | 319 | 68.9 |
| Part-time /Unemployed/Retired/Others | 96 | 20.7 |
| Students | 48 | 10.4 |
| Monthly income | | |
| 10,000 or below | 104 | 22.5 |
| 10,000 to 19,999 | 117 | 25.3 |
| 20,000 to 39,999 | 163 | 35.2 |
| 40,000 or above | 75 | 16.2 |
| Prefer not to say | 4 | 0.9 |

fewer number of regular sex partners [t(462) = -2.34, p < .05] and non-regular sex partners [t(462) = -2.73, p < .01], but more masturbation [χ2 = 11.81, p < .001] and viewing of pornographies [χ2 = 8.82, p < .01] during the COVID-19 period.

Among those who had sex with a regular, non-regular or casual sex partner during the COVID-19 period, 52.4%, 31.8% and 46.7% had condomless anal intercourse respectively. Compared to the pre-COVID period, participants tended to report more condomless anal intercourse with regular (52.4% vs 47.0%), non-regular (31.8% vs 25.3%), and casual (46.7% vs 25.0%) sex partners during the COVID-19 period. Among those who had sex with any sex partner during the COVID-19 period, more than one third (39.3%) had use drugs and alcohol during sex, and 14.2% had experienced group sex, such figures were lower than those reported in the pre-COVID period (i.e. 41.9% had use drugs and alcohol during sex, and 18.9% had experienced group sex).

## Descriptive statistics of potential factors of condomless anal intercourse during the COVID-19 period

The descriptive statistics of the studied potential factors to condomless anal intercourse sex are presented in Table 3. COVID-19 related individual factors were constituted of 1) perceived

**Table 2. Pattern and change in sexual behaviors among MSM before and during COVID-19 period.**

| | During COVID-19 | Before COVID-19 | Differences between groups | | |
|---|---|---|---|---|---|
| | Mean | SD | Mean | SD | |
| Number of regular sex partners (among all MSM) | 1.24 | 1.34 | 1.40 | 1.99 | t(462) = -2.34*a |
| Number of non-regular sex partners (among all MSM) | 2.09 | 4.58 | 2.64 | 5.28 | t(462) = -2.73**a |
| Number of casual sex partners (among all MSM) | 0.08 | 0.80 | 0.07 | 0.45 | n.s. a |
| | N | % | N | % | |
| Condomless anal intercourse with regular sex partner (among MSM who had sex with regular sex partner) | 182 | 52.4 | 155 | 47.0 | N.A. |
| Condomless anal intercourse with non-regular sex partner (among MSM who had sex with non-regular sex partner) | 69 | 31.8 | 61 | 25.3 | N.A. |
| Condomless anal intercourse with casual sex partner (among MSM who had sex with casual sex partner) | 7 | 46.7 | 4 | 25.0 | N.A. |
| Use of drugs and alcohol during sex (among MSM who had sex with any sex partner) | 157 | 39.3 | 164 | 41.9 | N.A. |
| Group sex (among MSM who had sex with any sex partner) | 57 | 14.2 | 74 | 18.9 | N.A. |
| Other sex behaviors (among all MSM) | | | | | |
| Masturbation | 222 | 47.9 | 199 | 43.0 | $\chi 2$ = 11.81***b |
| Use of sex toys | 20 | 4.3 | 14 | 3.0 | n.s. b |
| Virtual sex behavior (such as phone/camera) | 6 | 1.3 | 1 | 0.2 | n.s. b |
| Viewing pornographies | 242 | 52.3 | 225 | 48.6 | $\chi 2$ = 8.82**b |

Total number of MSM in the sample: 463

Total number of MSM who have sex with regular sex partner: 347 (during COVID-19) / 330 (before COVID-19)

Total number of MSM who have sex with non-regular sex partner: 217 (during COVID-19) / 341 (before COVID-19)

Total number of MSM who have sex with casual sex partner: 15 (during COVID-19) / 16 (before COVID-19)

Total number of MSM who have sex with any sex partner: 399 (during COVID-19) / 391 (before COVID-19)

[a] Within group difference tested by paired sample t-test

[b] Within group difference tested by Mcnemar test

*$p < .05$

**$p < .01$

***$p < .001$, n.s = non-significant, N.A. = No statistical significance tests were conducted due to different number of MSM who had sex before and during COVID-19 period

risk of COVID-19 infection in general; 2) perceived risk of COVID-19 infection during sexual intercourse; 3) perceived severity of the COVID-19; 4) COVID-19 risk reduction behaviors in general; and 5) COVID-19 risk reduction behaviors during sex encounters. For perceived risk of COVID-19 infection in general, over half of the participants had concerns with high chance of transmission through intimate contact behaviors (68.7%). On perceived risk of COVID-19 infection during sexual intercourse, more than one third (39.3%) raised concern of being infected with COVID-19 from a male non-regular partner. High severity of the COVID-19 was recognized by 44.1% of the participants. For general COVID-19 risk reduction behaviors, most of them reported washing hands (89.0%) and wearing masks (98.3%) frequently. For COVID-19 risk reduction behaviors during sex encounter, participants reported that they would avoid having sex with non-regular partner (43.7%), only have sex with one regular partner (45.7%), avoid group sex (67.9%), and wash hands before and after having sex (66.9%).

Participants also reported changes in contextual-level factors, including reduction in availability of sex venues (51.4%), places where sex with male partners is possible (74.3%), and entertainment venues visited by members (69.2%).

**Table 3. Descriptive statistics of potential factors of condomless anal intercourse with any sex partner among MSM during the COVID-19 period (N = 463).**

| | Mean | SD | N | % |
|---|---|---|---|---|
| **Individual factors** | | | | |
| 1. Perceived risk of COVID-19 infection in general | 3.66 | 0.95 | | |
| High chance of COVID-19 transmission through intimate contact behaviors (such as kissing/touching) (agree/strongly agree) | | | 318 | 68.7 [a] |
| Individuals with COVID-19 are highly contagious even they are asymptomatic (agree/strongly agree) | | | 220 | 47.5 [a] |
| 2. Perceived risk of COVID-19 infection during sexual intercourse | 2.69 | 1.13 | | |
| Concern of being infected with COVID-19 from a male regular partner (agree/strongly agree) | | | 77 | 16.6 [a] |
| Concern of being infected with COVID-19 from a male non-regular partner (agree/strongly agree) | | | 182 | 39.3 [a] |
| 3. Perceived severity of the COVID-19 (serious/very serious) | 3.31 | 1.12 | 204 | 44.1 [a] |
| 4. COVID-19 risk reduction behaviors in general | 3.58 | 0.67 | | |
| Stay at home (often/always) | | | 145 | 31.4 [a] |
| Wash hands (often/always) | | | 412 | 89.0 [a] |
| Wear masks (often/always) | | | 455 | 98.3 [a] |
| Measure temperature (often/always) | | | 190 | 41.1 [a] |
| Avoid inviting friends or relatives to one's home (often/always) | | | 178 | 38.5 [a] |
| Avoid going to crowded areas (such as bar/sauna) to find partners (often/always) | | | 240 | 53.8 [a] |
| 5. COVID-19 risk reduction behaviors during sex encounters | 2.63 | 0.73 | | |
| Avoid having same sex behaviors at all (often/always) | | | 48 | 11.1 [a] |
| Reduce sexual behaviors with partner (often/always) | | | 82 | 19.0 [a] |
| Avoid having sex with non-regular partner (often/always) | | | 174 | 43.7 [a] |
| Only have sex with one regular partner (often/always) | | | 190 | 45.7 [a] |
| Avoid group sex (often/always) | | | 266 | 67.9 [a] |
| Have sex behaviors with partner only at home (often/always) | | | 128 | 30.1 [a] |
| Have sex behaviors with partner only outside home (often/always) | | | 118 | 27.8 [a] |
| Wear a mask when having sex with partner (often/always) | | | 8 | 1.9 [a] |
| Reduce having physical contact (such as kiss/caress) when having sex (often/always) | | | 22 | 5.3 [a] |
| Wash hands before and after having sex (often/always) | | | 310 | 66.9 [a] |
| Sanitize surrounding areas after having sex (often/always) | | | 110 | 26.2 [a] |
| Understand whether the sex partner has COVID-19-related symptoms (often/always) | | | 133 | 31.7 [a] |
| Understand whether the sex partner has been abroad (often/always) | | | 119 | 25.7 [a] |
| Understand whether the sex partner has approached people infected with COVID-19 (often/always) | | | 101 | 21.8 [a] |

(*Continued*)

**Table 3.** (Continued)

| | Mean | SD | N | % |
|---|---|---|---|---|
| **Interpersonal factors** | | | | |
| 6. Social support | 5.66 | 1.86 | | |
| 7. Social isolation (≥6) | 1.55 | 0.56 | 165 | 35.6 [b] |
| 8. Condom negotiation | 4.58 | 0.76 | | |
| **Contextual-level factor** | | | | |
| 9. Collective efficacy | 3.00 | 1.04 | 134 | 28.9 [a] |
| 10. Change in sex-seeking environment | 2.13 | 0.69 | | |
| Number of venues where MSM can have sex with a man (reduced/greatly reduced) | | | 148 | 51.4 [c] |
| Number of gay-friendly venues (e.g., bars saunas) (reduced/greatly reduced) | | | 344 | 74.3 [c] |
| Number of social events available for MSM (reduced/greatly reduced) | | | 320 | 69.2 [c] |

[a] People who scored 4 or above (agree / strongly agree or often / always) on the respective item

[b] People who scored 6 or higher on social isolation

[c] People who scored 2 or below (reduced / greatly reduced) on the respective item

## Socio-ecological factors on condomless anal intercourse with any sex partners during the COVID-19 period

Results from the univariate logistic regression analyses reported that among all the socio-demographic factors, monthly income was significantly associated with condomless anal intercourse with any sex partners during the COVID-19 period (OR = 0.78, 95% CI = 0.64, 0.96, data not tabulated). Adjusted for monthly income, results from logistic regression analyses revealed that perceived severity of COVID-19 (aOR = 0.72, 95% CI = 0.58, 0.88), COVID-19 risk reduction behaviors in general (aOR = 0.68, 95% CI = 0.48, 0.96), COVID-19 risk reduction behaviors during sex encounters (aOR = 0.45, 95% CI = 0.30, 0.66), condom negotiation (aOR = 0.61, 95% CI = 0.44, 0.86) and collective efficacy (aOR = 0.79, 95% CI = 0.64, 0.98) were protective factors of condomless anal intercourse during the COVID-19 period (Table 4).

## Discussions

As the COVID-19 pandemic has great impact on sexual behavior of MSM, this study investigated change in the patterns of sex behavior, as well as examining the socio-ecological factors associated with condomless anal intercourse with any sex partners during the COVID-19 period among MSM in Hong Kong. The findings of the present study indicate that a high proportion of MSM have reduced their sexual risk during the COVID-19 period. In particular, compared to before the COVID-19 pandemic, there were significant decreases in the mean number of regular and non-regular sex partners among MSM during the COVID-19 period. A decrease in the proportion of MSM engaging in group sex was also observed during the COVID-19 period. Findings are consistent with previous studies that documented difficulties in finding sex partners [13] and a reduction in number of sex partners among MSM during the COVID-19 period [10–13]. In addition, consistent with previous findings [39], the present study also showed an increase in other solo sex activities which carry relatively little risk for HIV, such as masturbation and viewing pornographies. The decrease in number of sex partners and group sex, and increase in solo sex behaviors are likely a result of the restrictions in

**Table 4. Regression of socio-ecological factors on condomless anal intercourse with any partner among MSM during the COVID-19 period (among those who had sex with any sex partner in the past 6 months, N = 399).**

| | OR | 95% CI | aOR # | 95% CI |
|---|---|---|---|---|
| **Individual-level factors** | | | | |
| 1. Perceived risk of COVID-19 infection in general | 0.84 | [0.68, 1.03] | 0.82 | [0.65, 1.03] |
| 2. Perceived risk of COVID-19 infection during sexual intercourse | 0.80* | [0.67, 0.96] | 0.81 | [0.66, 1.00] |
| 3. Perceived severity of COVID-19 | 0.74*** | [0.61, 0.88] | 0.72*** | [0.58, 0.88] |
| 4. COVID-19 risk reduction behaviors in general | 0.62*** | [0.45, 0.85] | 0.68* | [0.48, 0.96] |
| 5. COVID-19 risk reduction behaviors during sex encounters | 0.50*** | [0.36, 0.69] | 0.45*** | [0.30, 0.66] |
| **Interpersonal factors** | | | | |
| 6. Social support | 0.92 | [0.82, 1.02] | 0.91 | [0.81, 1.02] |
| 7. Social isolation | 0.95 | [0.67, 1.36] | 0.86 | [0.56, 1.28] |
| 8. Condom negotiation during the COVID-19 period | 0.54*** | [0.40, 0.73] | 0.61*** | [0.44, 0.86] |
| **Contextual-level factor** | | | | |
| 9. Collective efficacy | 0.76*** | [0.62, 0.91] | 0.79* | [0.64, 0.98] |
| 10. Change in sex-seeking environment | 1.02 | [0.75, 1.36] | 0.92 | [0.65, 1.30] |

# adjusted for monthly income

*$p < .05$

**$p < .01$

***$p < .001$

social interactions brought by the extended COVID-19 social distancing measures implemented in Hong Kong.

Despite these promising changes, our findings raised some important concerns. Notably, among those who had a sex partner, an increase in condomless anal intercourse with all kinds of sex partners (i.e. regular sex partners, non-regular sex partners and casual sex partners) has been observed, suggesting that MSM might seek to engage in other risky sexual behaviors to compensate the difficulties in finding sex partners brought by the COVID-19 social restriction measures. More studies are needed to examine the overall impact of COVID-19 pandemic on sexual risk among MSM in Hong Kong.

Findings of the present study revealed that participants perceived a moderate level of risk of COVID-19 infection. For example, more than two third (68.7%) perceived a high chance of COVID-19 transmission through intimate contact behaviors, and more than one third (39.3%) had concern about being infected with COVID-19 from non-regular partner. Nearly half of the MSM perceived the threat of COVID-19 to be high. Results also showed that a significant number of MSM have been taking steps to reduce their risk of COVID-19 transmission by engaging in a range of COVID-19 risk reduction behaviors, such as wearing masks and washing hands frequently. Consistent with the literature [14, 32, 40], some of them even reported adopting COVID-19 precautionary behaviors with their sex partners during sex encounters, such as avoiding group sex, restricting sex with only regular partner and avoiding sex with non-regular partner, many of which are also considered effective in preventing HIV. Findings suggest that while MSM have been trying to minimize physical contact with their sex partners and modify their sex behaviors in order to reduce their HIV transmission, they may also utilize the similar strategies to reduce their COVID-19 transmission during the pandemic.

The present study also identified several important factors of condomless anal intercourse from various level. First, at the individual level, the present study indicated that several COVID-19 related factors, including perceived severity of COVID-19, and both COVID risk reduction behaviors in general and during sex encounters were protective factors to

condomless anal intercourse among MSM during the COVID-19 pandemic. This finding is similar to other studies, e.g. a study among male clients of female sex workers [20], investigating associations between risk perceptions regarding HIV and inconsistent condom use. It is plausible that those who are more concerned about COVID-19 are more health conscious and thus be more likely to adopt safe sex behaviors. Perceptions related to COVID-19 and COVID-19 risk reduction behavior could potentially be related to adoption of HIV preventive behaviors [41].

Second, it is interesting to note that while social support and loneliness are well-known social factors to sexual risk behaviors [22, 23], the present study failed to find a significant association between social support, social isolation and condomless anal intercourse during MSM during the COVID-19 pandemic. Instead, among the social factors, condom negotiation was the only significant factor of condomless anal intercourse in the present study. Those who engage in condom negotiation are more likely to build up trust and open communication with their partners, which are essential components for reaching an agreement to use a condom during sexual intercourse [26].

Finally, with regards to the contextual-level factors, findings were consistent with the literature that MSM with higher level of collective efficacy were less likely to report condomless anal intercourse [28, 29]. Those with higher level of collective efficacy are likely to believe that there is collective motivation to engage in safe sex behavior, thus reporting lower level of condomless anal intercourse during the COVID-19 pandemic.

## Implications of findings

Findings of our study demonstrated that MSM made significant changes to their sexual behavior during the COVID-19 pandemic. These include changes in the number of sex partners and the form of sexual activities, and forming new strategies to reduce their risks of infection from partners. They also expressed high levels of concern and perceived high levels of threats of COVID-19. We expect these changes to be important not only for reducing the COVID-19 transmission, but also for reducing HIV transmission. Despite the substantial changes in sexual behavior, we also noted concerns about the increased level of condomless anal intercourse with sex partners. More research is needed to understand how COVID-19 pandemic affects the sexual behavior of MSM in the longer term as the COVID-19 pandemic continues to reshape our social and sexual lives.

Regardless of the reduced sexual risk, we deemed that interventions on promoting safe sex during the COVID-19 pandemic are still needed and such interventions could emphasize prevention of both COVID-19 and HIV. As the Hong Kong government is gradually lifting up COVID-19 restrictions, interventions should be targeted to addressing the risk of COVID-19 under the new normal. Messages in reducing the risk of COVID-19 while having sex might also be particularly useful for reducing HIV among the MSM community.

The present study also found that that perceived severity of COVID-19 was associated with lower level of condomless anal intercourse. It would be crucial to promote realistic risk perceptions about COVID-19 by providing scientific information on the transmissibility and consequences of the disease. At the interpersonal level, interventions that promote open communications, and condom negotiation skills and strategies should be encouraged. The significant association between collective efficacy and lower levels of condomless anal intercourse also highlights the need for interventions to integrate community mobilization as part of HIV prevention. Such efforts should mobilize the MSM populations to address the challenges they face in HIV prevention. Promoting collection efficacy and mobilization can also address the stigma and discrimination experienced by MSM, which would be an important prerequisite for promoting safer sex among this population.

## Limitations

The present study was subject to several limitations. First, it was a cross-sectional study and thus causality cannot be assumed. Recall bias in sexual behaviors or other COVID-19 prevention behaviors might exist. Social desirability might also exist and some MSM might report higher level of COVID-19 preventive behaviors and lower level of risky sexual behaviors during the pandemic. There may also be selection bias as participants who chose to participate in the study might be those who were more concerned about COVID-19. Finally, due to the lack of validated measures on COVID-19 perceptions, most of the items were self-developed or adapted from previous studies, the psychometric properties of these items should be cautioned.

## Conclusion

The COVID-19 control measures have caused a dramatic impact on the sexual behavior of MSM in Hong Kong. As the Hong Kong government is gradually lifting up COVID-19 restrictions, it would be important to continue to monitor how changes in COVID-19 control measures and related perceptions would impact the HIV risk of MSM in Hong Kong. Interventions that promote condom use during the COVID-19 pandemic are still needed and such interventions could emphasize prevention of both COVID-19 and HIV.

## Author Contributions

**Conceptualization:** Phoenix K. H. Mo, Meiqi Xin, Zixin Wang, Joseph T. F. Lau.

**Formal analysis:** Phoenix K. H. Mo, Kam Hei Hui.

**Funding acquisition:** Phoenix K. H. Mo.

**Methodology:** Phoenix K. H. Mo.

**Project administration:** Fuk Yuen Yu, Ho Hin Lee.

**Supervision:** Phoenix K. H. Mo.

**Writing – original draft:** Phoenix K. H. Mo.

**Writing – review & editing:** Phoenix K. H. Mo, Xinchen Ye.

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
