## [Decision Letter · Decision Letter 0]

14 Nov 2023

PONE-D-23-30000Patterns of sex behaviors and factors associated with condomless sex during the COVID-19 pandemic among men who have sex with men in Hong KongPLOS ONE

Dear Dr. Mo,

Thank you for submitting your manuscript to PLOS ONE. After careful consideration, we feel that it has merit but does not fully meet PLOS ONE’s publication criteria as it currently stands. Therefore, we invite you to submit a revised version of the manuscript that addresses the points raised during the review process.

We look forward to receiving your revised manuscript.

Kind regards,

Nelsensius Klau Fauk, S.Fil., M., MHID, MSc, PhD

Academic Editor

PLOS ONE

Journal Requirements:

Reviewers' comments:

Reviewer's Responses to Questions

**Comments to the Author**

1. Is the manuscript technically sound, and do the data support the conclusions?

Reviewer #1: Yes

2. Has the statistical analysis been performed appropriately and rigorously? 

Reviewer #1: Yes

3. Have the authors made all data underlying the findings in their manuscript fully available?

Reviewer #1: No

4. Is the manuscript presented in an intelligible fashion and written in standard English?

Reviewer #1: Yes

5. Review Comments to the Author

Reviewer #1: Kindly find the attached comments on the “Patterns of sex behaviors and factors associated with condomless sex during the COVID-19 pandemic among men who have sex with men in Hong Kong”.

Abstract

- “The present study examined the patterns of sex behaviors before and during COVID-19, and identified the factors associated with condomless sex during COVID-19 from individual, interpersonal, and contextual level among men who have sex with men (MSM) in Hong Kong”. Here the authors stated that the participants were MSM. But later in the abstract they mentioned “However, a higher level of condomless sex with all types of sex partners during the COVID-19 period was also observed”. So, who were the study participants?

Introduction

- The introduction is too long. Be concise.

Methods

- The study design should be recognized from the title and abstract. But even after read the methods, I could not find the study design utilized by the authors.

- There is no information about the population and sampling, sampling techniques, and the calculation of the sample size.

- I calculated that at least 47 questions aside from the socio-demographic variables raised by the interviewers. They applied telephone surveys for the data collection. How the authors will explain the decision to use this approach instead of other methods of data collection?

- How did the authors explain the online methods they used to reach the participants? This approach was applied by a trained and experienced fieldworker and peer referral. How did a single person reach more than 600 participants? The authors should explain step-by-step process they used so that at the end they came up with 463 participants.

- In the introduction, the very first sentence is related to HIV. I highlighted and understand that “….COVID-19 can still be detected in semen and feces and persists even after the virus is no longer detected”. In the “Factors at the individual lever” and all other items, no more question about HIV. So the condomless sex is only focus on the COVID-19? Elaborate this.

Results

- In the inclusion criteria, the authors mentioned “self-reported having had sex with a male” or in other parts stated as MSM. But in the Table 1, they included homosexual, heterosexual, and bisexual. I am not specifically in this area, but I think the inclusion criteria need to be detailed.

- The data collection was conducted with telephone surveys. I am interested to know how did the authors/interviewers asked the monthly income? So that only 4 participants who “prefer not to say”.

- In the methods, the authors mentioned the measures they used: (1) Same-sex behaviors before and during the COVID-19 period, (2) Factors at the individual level, (3) Factors at the interpersonal level, and (4) Factors at the contextual level. Later, the authors divided the results into (1) Socio-demographic characteristics of the participants, (2) Pattern and changes of sexual behavior during the COVID-19 period, (3) Descriptive statistics of potential factors of CAI during the COVID-19 period, (4) Socio-ecological factors on CAI with any sex partners during the COVID-19 period. How did the authors retrieve the data in the results with the measurement they planned in the methods? I think the headings/sub-headings of the results should align with the domains they mentioned in the methods section.

- Under Table 3 (Descriptive statistics of potential….) the authors explain a, b, and c. What for are these in the analysis? They did not mention anything about the threshold scores in the methods.

- In the methods, the authors explained the “factors at the individual level: (1) Perceived risk of COVID-19 infection in general, (2) Perceived risk of COVID 19 infection during sexual intercourse, (3) Perceived severity of COVID-19, (4) COVID-19 risk reduction behaviors in general, and (5) COVID-19 risk reduction behaviors during sex. How did the authors get the data of “COVID-19 related worry” in the results? (line 273). No data about it in the Table 4.

Discussion

- Better to re-state the purpose of the study to start the discussion.

- The authors may analyze/discuss the differences between “perceived severity of COVID-19” with “perceived severity of HIV AIDS”. Find the article “Perceptions of determinants of condom use behaviors among male clients of female sex workers in Indonesia: A qualitative inquiry”

Overall:

- Do check the abbreviation policy of the journal. Is CAI a common abbreviation? I read many use UAI, CAS, CLAI, and many more for the similar matter.

6. PLOS authors have the option to publish the peer review history of their article (what does this mean?). If published, this will include your full peer review and any attached files.

Reviewer #1: No

---

## [Author Response · Author response to Decision Letter 0]

28 Dec 2023

Comments from the editor:

Response: Thank you for your comment. We've revised the manuscript to align with PLOS ONE's style requirements, including proper file naming.

Response: Thank you for your comment. We have incorporated the full ethics statement into the 'Methods' section of the manuscript. Additionally, we obtained verbal informed consent, and we've included this information in the statement.

Response: Thank you for your comment. We further cited an article as suggested by the reviewer to enrich the introduction section and the discussion section.

Fauk NK, Kustanti CY, Liana DS, Indriyawati N, Crutzen R, Mwanri L. Perceptions of Determinants of Condom Use Behaviors Among Male Clients of Female Sex Workers in Indonesia: A Qualitative Inquiry. Am J Mens Health. 2018 Jul;12(4):666-675. doi: 10.1177/1557988318756132. Epub 2018 Feb 22. PMID: 29468942; PMCID: PMC6131453.

Comments from the reviewer:

Abstract

- “The present study examined the patterns of sex behaviors before and during COVID-19, and identified the factors associated with condomless sex during COVID-19 from individual, interpersonal, and contextual level among men who have sex with men (MSM) in Hong Kong”. Here the authors stated that the participants were MSM. But later in the abstract they mentioned “However, a higher level of condomless sex with all types of sex partners during the COVID-19 period was also observed”. So, who were the study participants?

Response: Thank you for your comment. The participants of this study are MSM in Hong Kong. Types of sex partners mentioned in this study included regular sex partners, non-regular sex partners, and casual sex partners. We have clarified this in the manuscript.

Introduction

- The introduction is too long. Be concise.

Response: Thank you for your comment. We have revised the introduction section to make it be concise and to the point.

Methods

- The study design should be recognized from the title and abstract. But even after read the methods, I could not find the study design utilized by the authors.

Response: Thank you for your comment. This study is a cross-sectional study. We have revised the title, abstract and methods section to clarify this.

- There is no information about the population and sampling, sampling techniques, and the calculation of the sample size.

Response: Thank you for your comment. We have revised the methods section to acknowledge that “this study included a convenient sample”. We also included information about sample size calculation in the manuscript. 

- I calculated that at least 47 questions aside from the socio-demographic variables raised by the interviewers. They applied telephone surveys for the data collection. How the authors will explain the decision to use this approach instead of other methods of data collection? 

Response: Thank you for your comment. We have reviewed the methods section to justify the choice of telephone surveys.

- How did the authors explain the online methods they used to reach the participants? This approach was applied by a trained and experienced fieldworker and peer referral. How did a single person reach more than 600 participants? The authors should explain step-by-step process they used so that at the end they came up with 463 participants.

Response: Thank you for your comment. We have revised the methods section to provide further details regarding the recruitment procedure.

- In the introduction, the very first sentence is related to HIV. I highlighted and understand that “….COVID-19 can still be detected in semen and feces and persists even after the virus is no longer detected”. In the “Factors at the individual lever” and all other items, no more question about HIV. So the condomless sex is only focus on the COVID-19? Elaborate this. 

Response: Thank you for your comment. Since this study focused more on the influence of COVID-19 pandemic on sex behaviors, we focused on factors related to COVID-19. HIV-related factors were not investigated. We have revised the introduction section to clarify this.

Results

- In the inclusion criteria, the authors mentioned “self-reported having had sex with a male” or in other parts stated as MSM. But in the Table 1, they included homosexual, heterosexual, and bisexual. I am not specifically in this area, but I think the inclusion criteria need to be detailed. 

Response: Thank you for your comment. Since the target population of this study is MSM, self-reported having had sex with a male, which was more behaviorally defined, was one of the inclusion criteria. We acknowledged that men who choose to have sex with men are not necessarily homosexual. Therefore, we also measured sexual orientation, i.e. homosexual, heterosexual, and bisexual, as one of characteristics of participants. 

- The data collection was conducted with telephone surveys. I am interested to know how did the authors/interviewers asked the monthly income? So that only 4 participants who “prefer not to say”.

Response: Thank you for your comment. The participants were not requested to provide an accurate number of the monthly income. They were just requested to indicate a level, i.e. 10000 or below, 10000 to 19999, 20000 to 29999, and 40000 or above. We have revised the methods section to clarify this.

- In the methods, the authors mentioned the measures they used: (1) Same-sex behaviors before and during the COVID-19 period, (2) Factors at the individual level, (3) Factors at the interpersonal level, and (4) Factors at the contextual level. Later, the authors divided the results into (1) Socio-demographic characteristics of the participants, (2) Pattern and changes of sexual behavior during the COVID-19 period, (3) Descriptive statistics of potential factors of CAI during the COVID-19 period, (4) Socio-ecological factors on CAI with any sex partners during the COVID-19 period. How did the authors retrieve the data in the results with the measurement they planned in the methods? I think the headings/sub-headings of the results should align with the domains they mentioned in the methods section.

Response: Thank you for your comment. We have revised the headings to make sure that there is a match between the results section and the methods section.

- Under Table 3 (Descriptive statistics of potential….) the authors explain a, b, and c. What for are these in the analysis? They did not mention anything about the threshold scores in the methods.

Response: Since factors of condomless anal intercourse were rated on Likert Scale, we set thresholds to help us better describe characteristics of participants’ answers. We have revised the methods section to clarify the thresholds we used.

- In the methods, the authors explained the “factors at the individual level: (1) Perceived risk of COVID-19 infection in general, (2) Perceived risk of COVID 19 infection during sexual intercourse, (3) Perceived severity of COVID-19, (4) COVID-19 risk reduction behaviors in general, and (5) COVID-19 risk reduction behaviors during sex. How did the authors get the data of “COVID-19 related worry” in the results? (line 273). No data about it in the Table 4.

Response: Thank you for your comment. We did not include this information in our analysis. We have deleted it from the manuscript.

Discussion

- Better to re-state the purpose of the study to start the discussion.

Response: Thank you for your comment. We have revised the discussion section to re-state the purpose of the study prior to the commencement of the discussion.

- The authors may analyze/discuss the differences between “perceived severity of COVID-19” with “perceived severity of HIV AIDS”. Find the article “Perceptions of determinants of condom use behaviors among male clients of female sex workers in Indonesia: A qualitative inquiry”

Response: Thank you for your comment. We have revised the discussion section to compare our findings with studies investigating risk perceptions regarding HIV.

Overall:

- Do check the abbreviation policy of the journal. Is CAI a common abbreviation? I read many use UAI, CAS, CLAI, and many more for the similar matter.

Response: Thank you for your comment. We have revised the manuscript to replace CAI with condomless anal intercourse.

We would like to take this opportunity to thank the reviewer and the editor who have provided very constructive comments on our paper. Your comments have helped us strengthen our arguments and enhance the quality of our manuscript. We greatly appreciate your review of our resubmission and look forward to your reply in due course. Should you have any enquiries, feel free to contact us. Thank you very much for your consideration.

---

## [Decision Letter · Decision Letter 1]

1 Feb 2024

PONE-D-23-30000R1Patterns of sex behaviors and factors associated with condomless anal intercourse during the COVID-19 pandemic among men who have sex with men in Hong Kong: A cross-sectional studyPLOS ONE

Dear Dr. Mo,

Thank you for submitting your manuscript to PLOS ONE. After careful consideration, we feel that it has merit but does not fully meet PLOS ONE’s publication criteria as it currently stands. Therefore, we invite you to submit a revised version of the manuscript that addresses the points raised during the review process.

We look forward to receiving your revised manuscript.

Kind regards,

Nelsensius Klau Fauk, S.Fil., M., MHID, MSc, PhD

Academic Editor

PLOS ONE

Journal Requirements:

**Additional Editor Comments:**

Thanks to the authors for submitting the revised version which much improvement.

Please address a minor comment raised by the reviewer about the inconsistency of the numbers used in text and table in the results section.

Reviewers' comments:

Reviewer's Responses to Questions

**Comments to the Author**

1. If the authors have adequately addressed your comments raised in a previous round of review and you feel that this manuscript is now acceptable for publication, you may indicate that here to bypass the “Comments to the Author” section, enter your conflict of interest statement in the “Confidential to Editor” section, and submit your "Accept" recommendation.

Reviewer #1: All comments have been addressed

2. Is the manuscript technically sound, and do the data support the conclusions?

Reviewer #1: Yes

3. Has the statistical analysis been performed appropriately and rigorously? 

Reviewer #1: Yes

4. Have the authors made all data underlying the findings in their manuscript fully available?

Reviewer #1: Yes

5. Is the manuscript presented in an intelligible fashion and written in standard English?

Reviewer #1: Yes

6. Review Comments to the Author

Reviewer #1: Thank you for delivering the response from the authors.

I have read the new version and the response letter.

I attached the comments on the revised version of the paper.

Title

- It is clear now that the study is a cross-sectional study.

Abstract

- The new version provides a better understanding of the terminologies and the research methods.

Methods

- The new version has more detailed information on the procedure.

- Sample size calculation: need a reference/references

Results

- Need to be consistent with numbers after coma. i.e “Table 2. Participants reported an average of 1.2 regular partners (SD = 1.34), 2.09 non-regular 288 partners (SD = 4.58), and 0.08 casual partners (SD = 0.80) during the COVID-19 period”. In the table, they used 1.24; in the description, it is 1.2. Meanwhile, for numbers, they used two numbers after the coma.

I have no more comments. Generally, the new version is more explicit as various parts I was concerned about before have been detailed.

7. PLOS authors have the option to publish the peer review history of their article (what does this mean?). If published, this will include your full peer review and any attached files.

Reviewer #1: No

---

## [Author Response · Author response to Decision Letter 1]

4 Mar 2024

Comments from the editor:

Response: Thank you for your comment. We have reviewed the reference list. We slightly changed the information regarding reference No.1, No.5, and No.19 in the bibliography to make information provided correct. In addition, we deleted the reference No.9.

Thanks to the authors for submitting the revised version which much improvement. Please address a minor comment raised by the reviewer about the inconsistency of the numbers used in text and table in the results section. 

Response: Thank you for your comment. We have revised the manuscript according to the comment raised by the reviewer.

Comments from the reviewer:

Title

- It is clear now that the study is a cross-sectional study.

Response: Thank you for your comment in the previous round.

Abstract

- The new version provides a better understanding of the terminologies and the research methods.

Response: Thank you for your comment in the previous round.

Methods

- The new version has more detailed information on the procedure.

Response: Thank you for your comment in the previous round.

- Sample size calculation: need a reference/references

Response: Thank you for your comment. We have revised the manuscript to provide references for information regarding sample size calculation.

Results

- Need to be consistent with numbers after coma. i.e “Table 2. Participants reported an average of 1.2 regular partners (SD = 1.34), 2.09 non-regular 288 partners (SD = 4.58), and 0.08 casual partners (SD = 0.80) during the COVID-19 period”. In the table, they used 1.24; in the description, it is 1.2. Meanwhile, for numbers, they used two numbers after the coma.

Response: Thank you for your comment. We have revised the manuscript to make sure that numbers of the present study are rounded to 2 decimal places and percentages of the present study are rounded to 1 decimal place.

We would like to take this opportunity to thank the reviewer and the editor who have provided very constructive comments on our paper. Your comments have helped us strengthen our arguments and enhance the quality of our manuscript. We greatly appreciate your review of our resubmission and look forward to your reply in due course. Should you have any enquiries, feel free to contact us. Thank you very much for your consideration.

---

## [Editor Report · Decision Letter 2]

7 Mar 2024

Patterns of sex behaviors and factors associated with condomless anal intercourse during the COVID-19 pandemic among men who have sex with men in Hong Kong: A cross-sectional study

PONE-D-23-30000R2

Dear Dr. Mo,

We’re pleased to inform you that your manuscript has been judged scientifically suitable for publication and will be formally accepted for publication once it meets all outstanding technical requirements.

Kind regards,

Nelsensius Klau Fauk, S.Fil., M., MHID, MSc, PhD

Academic Editor

PLOS ONE
---

## [Editor Report · Acceptance letter]

26 Mar 2024

PONE-D-23-30000R2 

PLOS ONE

Dear Dr. Mo, 

I'm pleased to inform you that your manuscript has been deemed suitable for publication in PLOS ONE. Congratulations! Your manuscript is now being handed over to our production team.

Kind regards, 

on behalf of

Dr. Nelsensius Klau Fauk 

Academic Editor

PLOS ONE